# Variational Interaction Information Maximization for Cross-domain Disentanglement

**HyeongJoo Hwang**[1], **Geon-Hyeong Kim**[2], **Seunghoon Hong**[2], **Kee-Eung Kim**[1,2]
[1] Graduate School of AI, KAIST, Daejeon, Republic of Korea
[2] School of Computing, KAIST, Daejeon, Republic of Korea
{hjhwang, ghkim}@ai.kaist.ac.kr, {seunghoon.hong, kekim}@kaist.ac.kr

## Abstract

Cross-domain disentanglement is the problem of learning representations partitioned into domain-invariant and domain-specific representations, which is a key to successful domain transfer or measuring semantic distance between two domains. Grounded in information theory, we cast the simultaneous learning of domain-invariant and domain-specific representations as a joint objective of multiple information constraints, which does not require adversarial training or gradient reversal layers. We derive a tractable bound of the objective and propose a generative model named Interaction Information Auto-Encoder (IIAE). Our approach reveals insights on the desirable representation for cross-domain disentanglement and its connection to Variational Auto-Encoder (VAE). We demonstrate the validity of our model in the image-to-image translation and the cross-domain retrieval tasks. We further show that our model achieves the state-of-the-art performance in the zero-shot sketch based image retrieval task, even without external knowledge.

## 1 Introduction

There have been great interests in learning disentangled representation for various purposes, such as identifying sources of variation [3, 4, 15, 18, 20] for interpretability, obtaining representation invariant to nuisance factors [1, 8, 30, 34, 39, 40], and domain transfer [12, 26, 28, 35, 44, 47]. In particular, the cross-domain disentanglement problem [12] assumes the dataset composed of paired samples $(x \in X, y \in Y)$ where every sample has some shared information. The problem requires a model to learn a representation explicitly separated into three parts: domain-invariant representation shared across two data domains and domain-specific representations exclusive to each domain. This task is challenging since those representations must be (1) disentangled so that they are independent to one another, while (2) informative in such a way that every factor of variation is captured in the right part of the representation.

In recent studies, many models have been proposed to tackle important tasks related to cross-domain disentanglement, such as image-to-image translation [12, 26, 28, 35, 44] and Zero-Shot Sketch Based Image Retrieval (ZS-SBIR) [6, 9, 22, 23, 27, 38]. Although those models perform reasonably well with realistic datasets, most of them take a heuristic combination of techniques that regularize the latent space, such as cycle consistency loss [46], cross-reconstruction loss [27], adversarial training [14, 43], and Gradient Reversal Layer (GRL) [10]. Consequently, it is not obvious to interpret each module or identify key factors that contribute to disentanglement in their models.

In this paper, we address the cross-domain disentanglement problem with a novel principle based on information theory. Specifically, we train a generative model named *Interaction Information Auto-Encoder* (IIAE) whose representations are enforced to be informative but disentangled by information regularization terms that we will describe shortly. Leveraging representations learned by IIAE, we show that image manipulation tasks such as image translation and synthesis can be done in fine details. Furthermore, we demonstrate that IIAE outperforms Generative Adversarial Network (GAN) [13] based models in the cross-domain retrieval task. Lastly, we empirically show that our model

outperforms the state-of-the-art models for ZS-SBIR which strongly depend on external knowledge such as word embedding of class labels. Our contributions are three-fold:

1. We propose a novel information-theoretic framework to learn and disentangle shared and exclusive representations and derive a tractable lower bound of the optimization objective.

2. By bridging the lower bound of the objective and the Evidence Lower Bound (ELBO), we introduce IIAE, a simple and interpretable generative model trained by maximizing the lower bound.

3. The performance of IIAE are demonstrated on an extensive set of tasks, such as cross-domain image translation, cross-domain retrieval, and ZS-SBIR.

## 2   Method

Consider a set of paired data sampled from an unknown joint distribution $(x, y) \sim p_D(x, y)$, where each element of a pair $x \in X$ and $y \in Y$ is extracted from different domains $X$ and $Y$, respectively. We assume that two domains exhibit domain-specific factors of variations while sharing some common factors of variations. For instance, $x$ and $y$ can be images in different styles (*e.g.*, sketch and photo) sharing the same semantic content, or images of different content (*e.g.*, different types of car) sharing the same factors of variation (*e.g.*, rotation and scale).

Given this data, the goal of cross-domain disentanglement is to find the structured representation that can be factorized into three parts: domain-specific representations $Z^X$ and $Z^Y$ that capture the distinctive and exclusive characteristics of each domain $X$ and $Y$, respectively, and the shared representation $Z^S$ that captures common factors shared across the domains. Figure 1a describes our graphical model encoding this structure.

A typical way to learn a latent variable model is maximizing the marginal likelihood [21]. In our problem, we maximize the marginal likelihood of the joint distribution of $X$ and $Y$:

$$p_\theta(x, y) = \int dz^x dz^s dz^y p_{\theta_X}(x|z^x, z^s) p_{\theta_Y}(y|z^y, z^s) p(z^x) p(z^s) p(z^y), \tag{1}$$

where $\theta = \{\theta_X, \theta_Y\}$ denotes the parameter modeling the conditional distributions. Our objective is then training the generative model $p_\theta(x, y)$ that not only maximizes the joint distribution $p_D(x, y)$ by optimizing $\theta$, but also disentangles the exclusive representations $Z^X$ and $Z^Y$ from the shared representation $Z^S$. Below, we describe our approach to optimize the Eq. (1) while enforcing the disentanglement constraints on the latent representations.

### 2.1   Generative model for the joint distribution $p_D(x, y)$

Since the direct optimization of Eq. (1) is intractable, we employ variational inference based on Variational Auto-Encoder (VAE) [21]. Specifically, we approximate the true posterior distribution $p_\theta(z^x, z^s, z^y|x, y)$ using the approximated posterior $q_\phi(z^x, z^s, z^y|x, y)$, which is factorized according to the graphical model in Figure 1b as follows:

$$q_\phi(z^x, z^s, z^y|x, y) = q_{\phi_X}(z^x|x) q_{\phi_S}(z^s|x, y) q_{\phi_Y}(z^y|y), \tag{2}$$

where $q_{\phi_X}$ and $q_{\phi_Y}$ are encoders for domain-specific latent variable $Z^X$ and $Z^Y$, respectively, $q_{\phi_S}$ is the encoder for the shared latent variable $Z^S$, and $\phi = \{\phi_X, \phi_S, \phi_Y\}$ is the encoder parameter. In the following, we omit subscripts $\theta$ and $\phi$ for brevity. Using the Eq. (2), we can derive the ELBO of Eq. (1) as follow (see A.1 in the supplementary material for the derivation):

$$\log p(x, y) \geq \mathbb{E}_{q(z^x, z^s, z^y|x, y)} \left[ \log \frac{p(x, y, z^x, z^s, z^y)}{q(z^x, z^s, z^y|x, y)} \right] \tag{3}$$

$$= \mathbb{E}_{q(z^x|x) q(z^s|x, y)} [\log p(x|z^x, z^s)] + \mathbb{E}_{q(z^y|y) q(z^s|x, y)} [\log p(y|z^y, z^s)]$$
$$- D_{KL} [q(z^x|x) \| p(z^x)] - D_{KL} [q(z^y|y) \| p(z^y)]$$
$$- D_{KL} [q(z^s|x, y) \| p(z^s)]. \tag{4}$$

Unfortunately, maximizing the ELBO does not necessarily encourage the structured representations. This is mainly because we have no control over the assignment of the generative factors to representations learned by three different encoders $q(z^x|x)$, $q(z^s|x, y)$, and $q(z^y|y)$. Specifically, the following desiderata of the cross-domain disentanglement should be reflected in the objective:

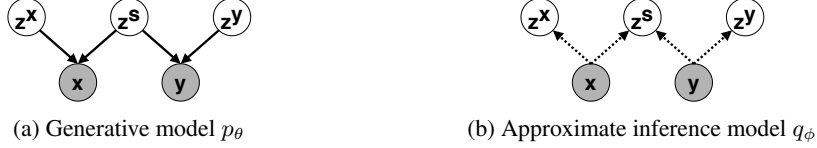

(a) Generative model $p_\theta$        (b) Approximate inference model $q_\phi$

Figure 1: Graphical models for cross-domain disentanglement.

1. **Disentanglement of $Z^X$, $Z^Y$ and $Z^S$**: the generative factors learned by $Z^X$, $Z^Y$ and $Z^S$ should be mutually exclusive to each other to avoid encoding redundant information.

2. **Decomposition of domain-specific and shared representations**: the generative factors exclusively presented in each domain should be captured by $Z^X$ and $Z^Y$, while the rest of factors shared across the domains should be encoded in $Z^S$.

To guide the model to learn desirable latent representations that satisfy the above properties, we propose to introduce regularizations on $q$ motivated by information theory, which are described below.

## 2.2 Information regularization on $q$ for cross-domain disentanglement

**Enforcing disentanglement**    Desirable shared and exclusive representations must be disentangled so that none of factors of variation is shared across any representations. Thus, we introduce regularizations that minimize the mutual information $I(Z^X; Z^S)$ and $I(Z^Y; Z^S)$ so that exclusive representations are statistically independent to shared representation, and vice versa. Here we only present our formulation for domain $X$, as the one for domain $Y$ is analogous.

To gain better insights on how minimizing the mutual information impacts the disentanglement, we rewrite $I(Z^X; Z^S)$ as follows (see A.2 in the supplementary for details):

$$I(Z^X; Z^S) = -I(X; Z^X, Z^S) + I(X; Z^X) + I(X; Z^S). \tag{5}$$

Surprisingly, Eq. (5) implies that minimizing the mutual information of $Z^X$ and $Z^S$ encourages them to be *jointly informative* to domain $X$ (the first term in RHS). Since the last two terms will penalize the total amount of information in $Z^X$ and $Z^S$, minimizing Eq. (5) will naturally encourage $Z^S$ and $Z^X$ to encode the *mutually exclusive* information of domain $X$.

However, we also notice that minimization of Eq. (5) does not enforce any constraints on separation of domain-specific and domain-invariant representation to $Z^X$ and $Z^S$; any arbitrary mutually exclusive factorization will be equally preferred, even those with no information captured in $Z^S$. It motivates us to introduce additional regularization to enforce a proper disentanglement on domain-specific and shared information.

**Enforcing decomposition**    To encourage decomposition of domain-specific and shared representation, we introduce a regularization on the shared latent variable $Z^S$. Specifically, we encourage $Z^S$ to capture the *shared* information across domains, which is enforced based on *interaction information* [31] (also known as co-information [2]).

Interaction information is a generalization of mutual information among three or more random variables, and quantifies the amount of shared information among them. Specifically, we define the interaction information among two domains $X$, $Y$ and the shared representation $Z^S$ as follows:

$$I(X; Y; Z^S) = I\left(X; Z^S\right) - I\left(X; Z^S|Y\right) \tag{6}$$
$$= I\left(Y; Z^S\right) - I\left(Y; Z^S|X\right), \tag{7}$$

where the equality in Eq. (7) holds due to symmetry. The above equations show how maximizing interaction information encourages $Z^S$ to encode the shared information. For instance, in Eq. (6), the first term in RHS is maximized when $Z^S$ becomes informative to $X$, while the second term will be minimized if such information in $Z^S$ can be *also inferred* from $Y$; the combination of both terms will naturally make $Z^S$ to encode information shared between $X$ and $Y$.

**Joint regularization**    Our final regularization on cross-domain disentanglement is obtained by combining regularizations on disentanglement and decomposition. To make analysis easier, we first present the objective with respect to domain $X$ and show the complete one on both domains later.

Combining Eq. (5) and (6), our preference for $q$ on domain $X$ (the negative of regularization) becomes

$$\max_q I(X;Y;Z^S) - I(Z^X;Z^S)$$

$$= \underbrace{I(X;Z^S) - I(X;Z^S|Y)}_{I(X;Y;Z^S)} + \underbrace{I(X;Z^X,Z^S) - I(X;Z^X) - I(X;Z^S)}_{-I(Z^X;Z^S)}$$

$$= I(X;Z^X,Z^S) - I(X;Z^X) - I(X;Z^S|Y). \tag{8}$$

**Optimization** Direct optimization of Eq. (8) is intractable since each term involves several intractable integrals. The details are in A.3 in the supplementary material.

The first term $I(X;Z^X,Z^S)$ in Eq. (8) is intractable since $q(x|z^x,z^s) = \frac{q(z^x,z^s|x)p_D(x)}{\int p_D(x,y) \ q(z^x,z^s|x,y) \ dxdy}$ involves intractable integral (unknown $p_D(x,y)$ and $p_D(x)$). Thus, we derive its lower bound with the generative distribution $p(x|z^x,z^s)$ as follows:

$$I(X;Z^X,Z^S) = \mathbb{E}_{q(z^x,z^s|x)p_D(x)}\left[\log \frac{q(x|z^x,z^s)}{p_D(x)}\right]$$

$$= H(X) + \mathbb{E}_{q(z^x,z^s|x)p_D(x)}\left[\log p(x|z^x,z^s)\right] + \mathbb{E}_{q(z^x,z^s)}\left[D_{KL}\left[q(x|z^x,z^s)\|p(x|z^x,z^s)\right]\right]$$

$$\geq H(X) + \mathbb{E}_{q(z^x,z^s|x)p_D(x)}\left[\log p(x|z^x,z^s)\right]$$

$$= H(X) + \mathbb{E}_{p_D(x,y) \ q(z^x|x) \ q(z^s|x,y)}\left[\log p(x|z^x,z^s)\right]. \tag{9}$$

Note that maximization of Eq. (9) not only maximizes $I(X;Z^X,Z^S)$ but also fits $p(x|z^x,z^s)$ to $q(x|z^x,z^s)$ so that we can utilize it as a decoder.

The second term $-I(X;Z^X)$ is intractable since $q(z^x) = \int p_D(x)q(z^s|x) \ dx$ is intractable (unknown distribution $p_D(x)$). We use $-\mathbb{E}_{p_D(x)}\left[D_{KL}\left[q(z^x|x)\|p(z^x)\right]\right]$ as its lower bound with the generative distribution $p(z^x)$ defined as the standard Gaussian, which is also known as the Variational Information Bottleneck (VIB) [1].

The last term is also intractable because $q(z^s|y) = \int p_D(x|y)q(z^s|x,y)dx$ is intractable (unknown $p_D(x|y)$). Similar to VIB, we use variational distribution $r^y(z^s|y)$ to maximize its lower bound:

$$-I(X;Z^S|Y) = -\mathbb{E}_{p_D(x,y)q(z^s|x,y)}\left[\log \frac{q(z^s|x,y)}{q(z^s|y)}\right]$$

$$= -\mathbb{E}_{p_D(x,y)q(z^s|x,y)}\left[\log \frac{q(z^s|x,y)r^y(z^s|y)}{r^y(z^s|y)q(z^s|y)}\right]$$

$$= -\mathbb{E}_{p_D(x,y)}\left[D_{KL}\left[q(z^s|x,y)\|r^y(z^s|y)\right]\right] + \mathbb{E}_{p_D(y)}\left[D_{KL}\left[q(z^s|y)\|r^y(z^s|y)\right]\right]$$

$$\geq -\mathbb{E}_{p_D(x,y)}\left[D_{KL}\left[q(z^s|x,y)\|r^y(z^s|y)\right]\right]. \tag{10}$$

Thus, the maximization of Eq. (10) not only minimizes $I(X;Z^S|Y)$ but also fits $r^y(z^s|y)$ to $q(z^s|y)$. Putting together, we are ready to derive the lower bound of the preference for $q$ on domain $X$ and $Y$:

$$\left(I(X;Y;Z^S) - I(Z^X;Z^S)\right) + \left(I(X;Y;Z^S) - I(Z^Y;Z^S)\right)$$

$$= 2 \cdot I(X;Y;Z^S) - I(Z^X;Z^S) - I(Z^Y;Z^S)$$

$$= I(X;Z^X,Z^S) + I(Y;Z^Y,Z^S) - I(X;Z^X) - I(Y;Z^Y) - I(X;Z^S|Y) - I(Y;Z^S|X)$$

$$\geq \mathbb{E}_{p_D(x,y)}\left[\ \mathbb{E}_{q(z^s|x,y)q(z^x|x)}\left[\log p(x|z^x,z^s)\right] + \mathbb{E}_{q(z^s|x,y)q(z^y|y)}\left[\log p(y|z^y,z^s)\right]\ \right]$$

$$\quad - \mathbb{E}_{p_D(x,y)}\left[\ D_{KL}\left[q(z^x|x)\|p(z^x)\right] + D_{KL}\left[q(z^y|y)\|p(z^y)\right]\ \right]$$

$$\quad - \mathbb{E}_{p_D(x,y)}\left[\ D_{KL}\left[q(z^s|x,y)\|r^y(z^s|y)\right] + D_{KL}\left[q(z^s|x,y)\|r^x(z^s|x)\right]\ \right]$$

$$\quad + H(X) + H(Y). \tag{11}$$

Surprisingly, many of the terms are also present in the ELBO. Thus, when we add the above lower bound to the ELBO objective to perform joint optimization, many of the terms above are obtained with very little additional cost by sharing parameters and computations, which we describe below.

## 2.3  Interaction Information Auto-Encoder

Our goal is to learn a latent variable model with maximum likelihood objective (ELBO in Eq. (3)) under the the information regularization for cross-domain disentanglement (Eq. (11)). Due to the

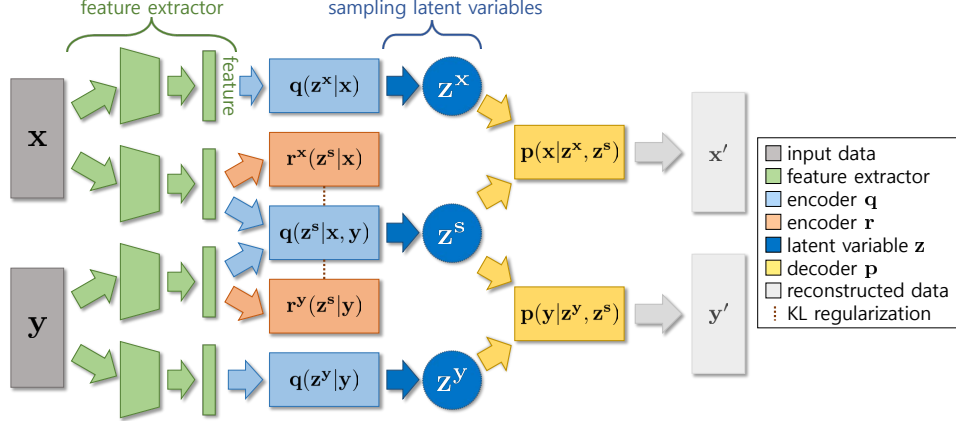

Figure 2: The architecture of Interaction Information Auto-Encoder.

difficulties in the constrained optimization, we relax this problem as a joint maximization problem similar to [34], which we name Interaction Information Auto-Encoder (IIAE) shown in figure 2, as follows (see A.3 in the supplementary material for details):

$$\max_{p,q} \mathbb{E}_{q(z^x, z^s, z^y, x, y)} \left[ \log \frac{p(x, y, z^x, z^s, z^y)}{q(z^x, z^s, z^y | x, y)} \right] + \lambda \left( 2 \cdot I(X;Y;Z^S) - I(Z^X; Z^S) - I(Z^Y; Z^S) \right)$$

$$\geq \max_{p,q,r} (1 + \lambda) \cdot \mathbb{E}_{p_D(x,y)} \left[ \, ELBO(p,q) \, \right]$$

$$+ \lambda \cdot \mathbb{E}_{p_D(x,y)} \left[ \, D_{KL} \left[ q(z^s | x, y) \| p(z^s) \right] \, \right] \tag{12}$$

$$- \lambda \cdot \mathbb{E}_{p_D(x,y)} \left[ \, D_{KL} \left[ q(z^s | x, y) \| r^y(z^s | y) \right] + D_{KL} \left[ q(z^s | x, y) \| r^x(z^s | x) \right] \, \right]. \tag{13}$$

This objective is essentially augmenting the ELBO with Eq. (12) and Eq. (13), which trades off the overall amount of information captured by the shared representation with that from the domain-specific information, by factor $\lambda$. This augmented term encourages the shared representation to exclude domain-specific factors of variation. Finally, note that Eq. (13) yields variational encoders $r^x(z^s | x)$ and $r^y(z^s | y)$ as byproducts of optimization, which is useful for many tasks such as image translation and retrieval where we need to extract the shared representation $z^s$ only from $x$ or $y$.

## 3 Related Work

**Invariant representation** Representation learning [25] focuses on feature extraction from the data that is informative to the given task. Information bottleneck (IB) [40] was introduced as an information theoretic regularization method to achieve minimal sufficient encoding by constraining the amount of information that latent variable encodes observed variable. IB enables the encoder to filter out nuisance factors and thus to generalize well. IB is later extended to deep VIB [1], which parameterizes IB with a neural network and optimizes the variational lower bound of the IB objective. VIB showed a close relationship to VAEs [21] and $\beta$-VAEs [15] by extending their models to unsupervised learning. Based on VIB, several methods were developed [34, 39] to learn encoders that capture only the factors of variation invariant to the given attribute. Similarly, a variant of VIB was proposed by [8] to learn a domain invariant representation by discarding domain specific variations. GRL [10] is another approach to achieve an invariant approach, which has been widely adopted to the tasks such as unlearning the bias in the input data [19], domain adaptation [10, 12], and zero-shot image retrieval [5]. The idea of learning invariant representations in zero-shot learning has been explored as well [6, 9, 22, 23, 27, 38], aiming to achieve domain-invariant representation by regularizing the model with multiple tasks or objectives.

**Disentangled representation** Based on $\beta$-VAEs [15], there has been extensive research on disentangled representation. Total correlation [42] is quantified as a measure of statistical dependency among all dimensions of the latent variable, which was the basis of the work by [3, 7, 11, 18, 20]. Modeling hierarchical structure in the latent space was also introduced by [16, 45], expecting that representations learned in each level is disentangled from other levels in the hierarchy. Extending the conditional generative models [17, 47], Cross-domain Disentanglement Networks (CdDN) [12]

introduced the concept of cross-domain disentanglement for image-to-image translation task, which is about disentangling domain-specific representation from the shared representation. As cross-domain disentanglement problem assumes paired dataset, there have been several follow-up studies [26, 28, 35, 44] that extend cross-domain disentanglement to the case only unpaired data is available.

## 4 Experiments

We employ experiments on image-to-image translation and image retrieval tasks to evaluate the quality of cross-domain disentanglement. In both tasks, the main objective is to evaluate how our method encodes the domain-specific and the shared information into different representations ($Z^X$, $Z^Y$, and $Z^S$).

### 4.1 Cross-domain Image Translation

**Datasets** We evaluate our method on two datasets: MNIST-CDCB [12] and Cars [36] datasets. In MNIST-CDCB [12] dataset, each pair $(x, y)$ consists of two images of the same digit but in different color patterns. Specifically, images in domain $X$ have color variations in the background, while the ones in domain $Y$ have variations in the foreground.We use 50,000 / 10,000 pairs of train/test samples following [24]. Cars [36] is a dataset of car CAD images with equally spaced variations in orientation, 4 different angles in pitch and 24 in yaw. We employ 92 pairs of $(x, y)$ per a car, where $x$ is fixed as a frontal view of every pitch, and $y$ is rotated view of rest 23 different angles in yaw. Out of those 16,836 pairs of 183 cars, we assigned 16,192 pairs of 176 cars to train set and 644 pairs of 7 cars to test set.

**Method** Translating an image across domains ($X \to Y$ or $Y \to X$) can be done naturally by our method. Specifically, we translate image $x$ in domain $X$ to domain $Y$ by (1) extracting its shared representation using the mean $\mu_x^s$ of $r^x(z^s|x)$, (2) sampling the domain-specific representation from the prior $z^y \sim p(z^y)$, and (3) generating the image by the mean $y'$ of $p(y|\mu_x^s, z^y)$. When we have the reference image $y$ in another domain, we can also conduct a guided translation by replacing the second step to extract domain-specific representation using the $z^y = \mu^y$, the mean of $q(z^y|y)$. Note that translation in the opposite direction can be done similarly. For network architecture, we employ the settings used in [12] with some minor modifications. We leave all the implementation details and hyperparameter settings in D in the supplementary material.

**Results** Table 1 shows the result of image translation with IIAE. For each row, we show the ground-truth pair $(x, y)$ (the first and sixth column), and the translated images between the domains. We present two types of translation results obtained by (1) sampling domain-specific representation from the prior $z \sim p(z)$ (columns 2∼4 and 7∼9) and (2) using the one extracted from the ground-truth pair $\mu \sim r(\cdot)$ (columns 5 and 10). More results can be found in C.1 in the supplementary material.

In MNIST-CDCB [12] dataset (upper half), we observe that shared and exclusive representations learned by IIAE are disentangled in a way that shared representation encoders only preserve the shape information and domain-specific encoders capture only the color information. We also observe multi-modal outputs in the translation results, which implies that various generative factors exclusively presented in each domain are captured by domain-specific representation. Furthermore, the images of the first and the last columns look alike as well as fifth column and sixth column do, which tells us that our shared representation encoders $r^x(z^s|x)$ and $r^y(z^s|y)$ provide nicely aligned representation.

In Cars [36] dataset (bottom half), cross-domain disentanglement is much more challenging since the object in each training pair $(x, y)$ can have different geometric configurations. From these data, the model should learn that the shared representation is the car identity, and the domain-specific variations are about the types of geometric transformations (fixed to front-view in domain $X$ and different rotation angles in domains $Y$'s). Under those challenges, the results show that IIAE can successfully learn disentangled representations. When translating an image from $X$ to $Y$ domain, it generates various orientations while keeping the car identity (second to fourth columns), whereas producing the consistent front-view images when translated in reverse direction (seventh to tenth columns). Quantitative evaluation on the sample generation is in the supplementary material B.1.

### 4.2 Image Retrieval

For quantitative evaluation of cross-domain disentanglement, we apply our method to the task of image retrieval. Given a query image, the objective is to find its nearest neighbors from the database

| $X \rightarrow Y$ | | | | $Y \rightarrow X$ | | | |
| --- | --- | --- | --- | --- | --- | --- | --- |
| **Input** | **Outputs w/ different $z^y$** | | | **Input** | **Outputs w/ different $z^x$** | | |
| x | $z_1^y, z_2^y, z_3^y \sim p(z^y)$ | | $\mu^y$ | y | $z_1^x, z_2^x, z_3^x \sim p(z^x)$ | | $\mu^x$ |

Table 1: Cross-domain translation results in MNIST-CDCB [12] (top half) and Cars [36] (bottom half) generated by IIAE. In MNIST-CDCB, domain-specific factors are color variation in the background ($X$) and the foreground ($Y$) while the common factor is the digit identity. In Cars, domain-specific factors only exists in $Y$, views in 23 different yaw angles, while the ones in $X$ is fixed to the front-view. The shared factor is the car identity.

images, where the query and images in a database are from different domains exhibiting some exclusive characteristics. The main challenge in this task is to learn image representation *invariant* to domain-specific characteristics, such that the distance between the query and database image in a representation space is aligned with their semantic similarity.

We address this task by exploiting the *shared* representation learned by our method. Given a query image $x \in X$ and the one from a database $y \in Y$, we compute their similarity by (1) extracting the shared representations independently by the mean $\mu_x^s$ of $r^x(z^s|x)$ and the mean $\mu_y^s$ of $r^y(z^s|y)$ and (2) computing their distance by $d(\mu_x^s, \mu_y^s)$ with a distance metric $d$ (*e.g.*, Euclidean distance, cosine distance, *etc.*). Then the retrieval is performed by extracting the $K$-nearest neighbors in the database.

### 4.2.1 Cross-domain retrieval

**Datasets**  We tested our model with MNIST-CDCB [12], Facades [41], and Maps [17] datasets. In Facades [41] dataset, each pair $(x, y)$ is made up of an image of semantic label map and photo of the same building. We use 400 / 100 / 106 pairs of train/valid/test samples following [41]. In Maps [17] dataset, each pair $(x, y)$ is composed of an image of map and a satellite image of the same area. We use 1096 / 1098 pairs of train/test samples following [17].

**Results**  Following [12], we compute the nearest neighbor using the Euclidean distance and evaluate the performance by the Recall@1 metric.[1] We compare our method with two baselines, CdDN [12] and DRIT [26], each of which is one of the most representative image to image translation models that encourage the cross-domain disentanglement in the representation with paired and unpaired dataset respectively. In order to make a fair comparison, we re-trained DRIT using the paired data via minor modification to the author's code to take advantage of the paired data. Table 2 summarizes

Table 2: Shared (exclusive) representation based retrieval on MNIST-CDCB [12], Maps [17], and Facades [41] dataset. CD/CB stand for colored digit/background, S/M stand for satellite/map, and F/L stand for facade/label respectively.

| Dataset | MNIST-CDCB | | Maps | | Facades | |
|---|---|---|---|---|---|---|
| Models | CD $\rightarrow$ CB | CB $\rightarrow$ CD | S $\rightarrow$ M | M $\rightarrow$ S | F $\rightarrow$ L | L $\rightarrow$ F |
| DRIT [26] | - | - | 33.8 (0.09) | 37.3 (0.09) | 31.1 (0.94) | 44.3 (0.94) |
| CdDN [12] | 99.6 (0.0) | 99.6 (0.0) | 91.4 (0.18) | 96.9 (0.09) | 84.9 (0.94) | 89.6 (0.0) |
| IIAE | **99.7** (0.01) | **99.7** (0.01) | **96.6** (0.09) | **97.3** (0.0) | **96.2** (0.94) | **99.1** (0.94) |

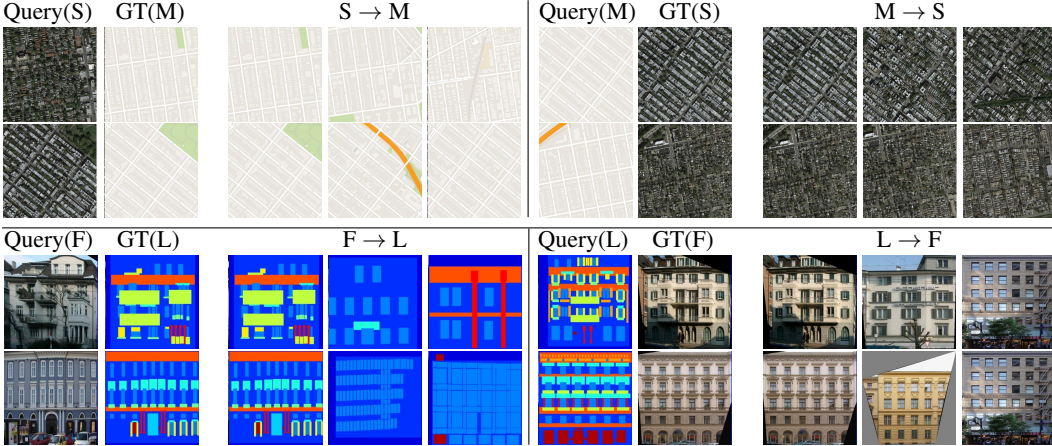

Figure 3: Qualitative examples of cross-domain retrieval (Top-1,2,3 from the left to the right) in Facades and Maps using IIAE. GT stands for ground truth.

the result of cross-domain retrieval with MNIST-CDCB, Maps, and Facades datasets. Evaluation of DRIT on MNIST-CDCB was intractable because the dimensionality of the shared representation as well as the size of the test set were too large. In MNIST-CDCB, both IIAE and CdDN both perform almost perfectly. This might be because the ground truth factors of variation inherent in the dataset is simple. However, in Maps [17] and Facades [41] datasets, we observe that IIAE outperforms all the baselines in any direction of the retrieval exhibiting well balanced performance in two directions. On the other hand, CdDN shows relatively poor performance on satellite→map in Maps and facade→label in Facades, and DRIT shows the worst performance in Facades and Maps, implying that the learned latent representations of two data domains are not aligned well. This shows that IIAE is more successful in capturing the complex factors of variation that are present in more realistic datasets such as Maps and Facades. Figure 3 presents the examples of top-3 images retrieved by IIAE in Maps (top two rows) and Facades (bottom two rows). All of top-1 images in figure 3 are the ground truth of the query. Furthermore, it is remarkable that most of images retrieved as second or third closest ones also have geometrical structure similar to the query image. Additional qualitative results of the retrieval can be found in supplementary material C.2.

**Ablation study** We also conducted cross-domain retrieval with *domain-specific representations* as an ablation study. The results are summarized in Table 2 with parenthesized numbers. We observe that the retrieval accuracy approaches near zero, which indicates that the learned domain-specific representations encode information only presented in each domain, as desired.

### 4.2.2 Zero-shot sketch based image retrieval (ZS-SBIR)

**Dataset** ZS-SBIR [22] is an extension of sketch based image retrieval task where none of the classes in the test set is exposed when training a retrieval model. We evaluate our model on Sketchy (Extended) [29, 37], one of the most widely used datasets of sketch and photo images in sketch-based image retrieval (SBIR) task. We employ the extended version of Sketcy dataset (Sketcy Extended) [29], which is composed of *un-aligned* images of 73,002 photos and 75,479 sketches distributed in 125 different classes. To learn our model without ground-truth pairs, we randomly sample one sketch and one photo per category to pair up one training sample. The factors of variation shared across two

Table 3: Evaluation on the Sketchy Extended dataset [29, 37]. WordEmb stands for word embedding.

| Models | Feature Dimension | Evaluation metric | | External knowledge | | |
|---|---|---|---|---|---|---|
| | | mAP | P@100 | Attribute | WordEmb. | WordNet [33] |
| SAE [23] | 300 | 0.216 | 0.293 | ✓ | ✓ | - |
| FRWGAN [9] | 512 | 0.127 | 0.169 | ✓ | - | - |
| ZSIH [38] | 64 | 0.258 | 0.342 | - | ✓ | - |
| CAAE [22] | 4096 | 0.196 | 0.284 | - | - | - |
| SEM-PCYC [6] | 64 | 0.349 | 0.463 | - | ✓ | ✓ |
| LCALE [27] | 64 | 0.476 | 0.583 | - | ✓ | - |
| IIAE | 64 | **0.573** | **0.659** | - | - | - |

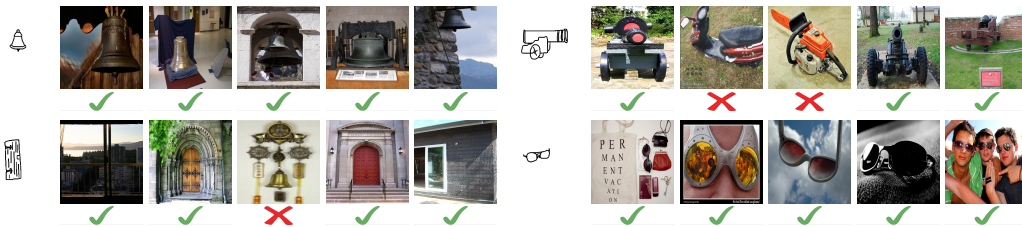

Figure 4: Top-5 ZS-SBIR samples from IIAE on the Sketchy Extended dataset. Sketches in the first and seventh columns are queries and rest are retrieved candidates (Top-1 to 5 from the left to the right). Green checkmark indicates correct retrieval, whereas red crossmark indicates wrong retrieval.

domains are the class of the object, while the exclusive ones are scale, translation, orientation, and style in both domains. We used train / test splits (100/25 categories) same as [6, 27] and extracted features of images from VGG16 and finetuned with the train set of Sketchy Extended. Those extracted features are used as input to IIAE.

**Results** We conducted the retrieval using cosine similarities, as used by [22], between shared representations extracted from IIAE. We compare IIAE with various baselines, SAE [23], FRWGAN [9], ZSIH [38], CAAE [22], SEM-PCYC [6], and LCALE [27], which are designed for ZS-SBIR or general zero shot learning. Following the previous works [6, 38], we chose mean average precision (mAP) and Precision@100 (P@100) as evaluation metric. Table 3 summarizes the result. It shows that IIAE outperforms all competitive methods, although some of them are specialized to this task and exploit side information such as attribute information of image, word embedding[32], or WordNet [33]. The result implies that IIAE successfully learns to associate semantic structure of sketches and images while being generalized well to unseen classes, which can be explained by two different information constraints on the shared representation; Eq. (13) enforces $Z^S$ to discard domain specific information while Eq. (4) encourages $Z^S$ to be a minimal sufficient statistic so that it generalizes well to unseen classes. Note that we can control the balance between being invariant and being compressive with $\lambda$. We also evaluated the effect of terms in the IIAE objective as an ablation study in the supplementary material B.3. Figure 4 shows the qualitative result of ZS-SBIR. It is notable that even the incorrectly retrieved images in figure 4 have visual or semantic correspondence to their query images. For instance, given a sketch of cannon as a query, a motorcycle and a saw are wrongly retrieved by IIAE, but the motorcycle is semantically relevant to the cannon due to its wheels whereas the saw is visually close to the motorcycle. Similarly, an image of bells is falsely retrieved by a sketch of door due to their visual similarity. Additional visualization of the ZS-SBIR results is in the supplementary material C.3.

## 5 Conclusion

In this paper, we investigate an approach for cross-domain disentanglement. The proposed approach, coined Interaction Information Auto-Encoder, extends the VAE with a novel regularization inspired by information theory, which are principled, interpretable, and nicely integrated into ELBO objective to encourage disentanglement of domain-specific and shared representations. The effectiveness of the proposed method is demonstrated on multiple applications, such as image-to-image translation and image retrieval.

## Broader Impact

Our method provides an information theoretic perspective on representation learning, and is likely to accelerate research in areas that involve datasets with two data domains with some common factors of variation. One of such areas is image to image translation we tackled in this paper. Beyond the image translation task, our method could be potentially applied to NLP tasks, such as language translation or text summarization where the source and the target data domains share semantics while they also have domain specific factors of variation in syntax. Leveraging IIAE, one could transform a sample from one domain to the other and measure the semantic similarity between languages from two different domains. However, one may exploit disentangled representations for wrongful purposes. For example, our approach could be adopted for Deepfake to generate more diverse fake images. Lastly, we do not see any serious consequences of system failure.

## Acknowledgments and Disclosure of Funding

This work was supported by the National Research Foundation (NRF) of Korea (NRF-2019R1A2C1087634 and NRF-2019M3F2A1072238), the Ministry of Science and Information communication Technology (MSIT) of Korea (IITP No. 2020-0-00940, IITP No. 2019-0-00075, IITP No. 2017-0-01779, IITP No. 2020-0-00153, and IITP No. 2016-0-00464), the ETRI (Contract No. 20ZS1100), and Samsung Electronics.

## Footnotes

[1]In the MNIST-CDCB[12] dataset, we only count the ground-truth pair of the query as a hit, whereas in [12] any retrieved image containing the same digit as a hit, which is why the scores are lower than originally reported. In the Facades [41] dataset, we present the results on the *test set*, while the results on the validation set is reported in [12]. We also report the result on the validation set in the supplementary material B.2.

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
