[Supplementary Material]

# Appendices

## A  Proofs

### A.1  Evidence Lower Bound on $p_\theta(x, y)$

$$\log p(x, y) = \log \int p(x|z^x, z^s)p(y|z^y, z^s) \ p(z^x)p(z^s)p(z^y) \ dz^x dz^s dz^y$$

$$= \log \int \frac{p(x|z^x, z^s)p(y|z^y, z^s) \ p(z^x)p(z^s)p(z^y)}{q(z^x|x)q(z^s|x, y)q(z^y|y)} \ q(z^x|x)q(z^s|x, y)q(z^y|y) \ dz^x dz^s dz^y$$

$$= \log \mathbb{E}_{q(z^x|x)q(z^s|x,y)q(z^y|y)} \left[ \frac{p(x|z^x, z^s)p(y|z^y, z^s) \ p(z^x)p(z^s)p(z^y)}{q(z^x|x)q(z^s|x, y)q(z^y|y)} \right]$$

$$\geq \mathbb{E}_{q(z^x|x)q(z^s|x,y)q(z^y|y)} \left[ \log \frac{p(x|z^x, z^s)p(y|z^y, z^s) \ p(z^x)p(z^s)p(z^y)}{q(z^x|x)q(z^s|x, y)q(z^y|y)} \right]$$

$$= \mathbb{E}_{q(z^x|x)q(z^s|x,y)} \left[ \log p(x|z^x, z^s) \right] + \mathbb{E}_{(z^s|x,y)q(z^y|y)} \left[ \log p(y|z^y, z^s) \right]$$

$$- D_{KL} \left[ q(z^x|x) \| p(z^x) \right] - D_{KL} \left[ q(z^s|x, y) \| p(z^s) \right] - D_{KL} \left[ q(z^y|y) \| p(z^y) \right].$$

### A.2  $I(Z^X; Z^S) = I(X; Z^X) + I(X; Z^S) - I(X; Z^X, Z^S)$.

Interaction information [31] between three random variables is defined as follows.

$$I(X; Y; Z) = I(X; Y) - I(X; Y|Z) = I(X; Z) - I(X; Z|Y) = I(Y; Z) - I(Y; Z|X). \tag{14}$$

Using the last equality, we obtain the following expression of mutual information between $Z^X$ and $Z^Y$:

$$I(Z^X; Z^S) = I(Z^X; X) - I(Z^X; X|Z^S) + I(Z^X; Z^S|X). \tag{15}$$

Due to the structural assumption on $q$, $q(z^x|x) = q(z^x|x, z^s)$ holds. Thus, the last term in the above equation disappears:

$$I(Z^X; Z^S|X) = H(Z^X|X) - H(Z^X|X, Z^S) = H(Z^X|X) - H(Z^X|X) = 0,$$

which yields

$$I(Z^X; Z^S) = I(X; Z^X) - I(X; Z^X|Z^S)$$
$$= I(X; Z^X) + I(X; Z^S) - I(X; Z^X, Z^S). \tag{16}$$

### A.3  Derivation of full objective

#### A.3.1  Lower bound on $(I(X; Y; Z^S) - I(Z^X; Z^S)) + (I(X; Y; Z^S) - I(Z^Y; Z^S))$

Here we derive the lower bound on $I(X; Y; Z^S) - I(Z^X; Z^S)$ since the one on $I(X; Y; Z^S) - I(Z^Y; Z^S)$ is analogous.

$$I(X; Y; Z^S) - I(Z^X; Z^S)$$
$$= \left( \cancel{I(X; Z^S)} - I(X; Z^S|Y) \right) - \left( I(X; Z^X) + \cancel{I(X; Z^S)} - I(X; Z^X, Z^S) \right)$$
$$= I(X; Z^X, Z^S) - I(X; Z^X) - I(X; Z^S|Y) \tag{17}$$
$$= H(X) + \mathbb{E}_{p_D(x)q(z^s, z^x|x)} \left[ \log q(x|z^x, z^s) \right] - \mathbb{E}_{p_D(x)} \left[ D_{KL} \left[ q(z^x|x) \| q(z^x) \right] \right]$$
$$- \mathbb{E}_{p_D(x,y)} \left[ D_{KL} \left[ q(z^s|x, y) \| q(z^s|y) \right] \right]$$

where $q(x|z^x, z^s) = \frac{q(z^x, z^s|x)p_D(x)}{\int p_D(x,y) \ q(z^x, z^s|x,y) \ dxdy}$, $q(z^x) = \int q(z^x|x)p_D(x)dx$,
and $q(z^y) = \int q(z^y|y)p_D(y)dy$ require intractable integrals. Thus, we need to derive the lower bound on Eq. (17).

**Variational lower bound on** $I(X; Z^X, Z^S)$ :

Note that $q(z^x, z^s)$ is intractable due to the unknown density of $p_D(x, y)$: $q(z^x, z^s) = \int p_D(x, y) \, q(z^x, z^s | x, y) \, dx \, dy$

Consequently, $q(x | z^x, z^s) = \frac{q(z^x, z^s | x) p_D(x)}{q(z^x, z^s)}$ is intractable. Thus, we would like to bring the generative distribution $p(x | z^x, z^s)$ to derive a lower bound such that:

$$
\begin{aligned}
I(X; Z^X, Z^S) &= \mathbb{E}_{q(z^x, z^s | x) p_D(x)} \left[ \log \frac{q(x | z^x, z^s)}{p_D(x)} \right] \\
&= H(X) + \mathbb{E}_{q(z^x, z^s | x) p_D(x)} \left[ \log q(x | z^x, z^s) - \log p(x | z^x, z^s) + \log p(x | z^x, z^s) \right] \\
&= H(X) + \mathbb{E}_{q(z^x, z^s | x) p_D(x)} \left[ \log p(x | z^x, z^s) \right] + \mathbb{E}_{q(z^x, z^s)} \left[ D_{KL} \left[ q(x | z^x, z^s) \| p(x | z^x, z^s) \right] \right] \\
&\geq H(X) + \mathbb{E}_{q(z^x, z^s | x) p_D(x)} \left[ \log p(x | z^x, z^s) \right] \\
&= H(X) + \int q(z^x, z^s | x) p_D(x) \, \log p(x | z^x, z^s) \, dx \, dz^x \, dz^s \\
&= H(X) + \int p_D(x) \left( \int q(z^x, z^s | x, y) \, p_D(y | x) \, dy \right) \log p(x | z^x, z^s) \, dx \, dz^x \, dz^s \\
&= H(X) + \int p_D(x) \, q(z^x | x) \left( \int q(z^s | x, y) \, p_D(y | x) \, dy \right) \log p(x | z^x, z^s) \, dx \, dz^x \, dz^s \\
&= H(X) + \int p_D(x, y) \, q(z^x | x) \, q(z^s | x, y) \log p(x | z^x, z^s) \, dx \, dy \, dz^x \, dz^s \\
&= H(X) + \mathbb{E}_{p_D(x, y) \, q(z^x | x) \, q(z^s | x, y)} \left[ \log p(x | z^x, z^s) \right]
\end{aligned}
$$

Thus, maximization of $\mathbb{E}_{p_D(x, y) \, q(z^x | x) \, q(z^s | x, y)} \left[ \log p(x | z^x, z^s) \right]$ not only maximizes $I(X; Z^X, Z^S)$, but also fits $p(x | z^x, z^s)$ to $q(x | z^x, z^s)$, so that we can utilize it as a decoder.

**Variational upper bound on** $I(X; Z^S | Y)$ :

Note that $q(z^s | y) = \int p_D(x | y) \, q(z^s | x, y) \, dx$ is intractable. Thus,

$$
\begin{aligned}
I(X; Z^S | Y) &= \mathbb{E}_{p_D(x, y) q(z^s | x, y)} \left[ \log \frac{q(z^s | x, y)}{q(z^s | y)} \right] && \left( = \mathbb{E}_{p_D(x, y)} \left[ D_{KL} \left[ q(z^s | x, y) \| q(z^s | y) \right] \right] \right) \\
&= \mathbb{E}_{p_D(x, y) q(z^s | x, y)} \left[ \log \frac{q(z^s | x, y) r^y(z^s | y)}{r^y(z^s | y) q(z^s | y)} \right] \\
&= \mathbb{E}_{p_D(x, y)} \left[ D_{KL} \left[ q(z^s | x, y) \| r^y(z^s | y) \right] \right] - \mathbb{E}_{p_D(y)} \left[ D_{KL} \left[ q(z^s | y) \| r^y(z^s | y) \right] \right] \\
&\leq \mathbb{E}_{p_D(x, y)} \left[ D_{KL} \left[ q(z^s | x, y) \| r^y(z^s | y) \right] \right] && (18)
\end{aligned}
$$

Thus, minimization of $\mathbb{E}_{p_D(x, y)} \left[ D_{KL} \left[ q(z^s | x, y) \| r^y(z^s | y) \right] \right]$ not only minimizes $I(X; Z^S | Y)$, but also fits $r^y(z^s | y)$ to $q(z^s | y)$.

**Variational upper bound on** $I(X; Z^X)$ :

Similar to Eq. (18), $I(X; Z^X) = \mathbb{E}_{p_D(x)} \left[ D_{KL} \left[ q(z^x | x) \| q(z^x) \right] \right] \leq \mathbb{E}_{p_D(x)} \left[ D_{KL} \left[ q(z^x | x) \| p(z^x) \right] \right]$.

**Overall information preference** :

Putting together, we can derive the lower bound of the preference for $q$ on domain $X$ and $Y$:

$$
\begin{aligned}
& (I(X; Y; Z^S) - I(Z^X; Z^S)) + (I(X; Y; Z^S) - I(Z^Y; Z^S)) \\
&= 2 \cdot I(X; Y; Z^S) - I(Z^X; Z^S) - I(Z^Y; Z^S) \\
&= I(X; Z^X, Z^S) + I(Y; Z^Y, Z^S) - I(X; Z^X) - I(Y; Z^Y) - I(X; Z^S | Y) - I(Y; Z^S | X) \\
&\geq \mathbb{E}_{p_D(x, y)} \left[ \mathbb{E}_{q(z^s | x, y) q(z^x | x)} \left[ \log p(x | z^x, z^s) \right] + \mathbb{E}_{q(z^s | x, y) q(z^y | y)} \left[ \log p(y | z^y, z^s) \right] \right] && (19) \\
& \quad - \mathbb{E}_{p_D(x, y)} \left[ D_{KL} \left[ q(z^x | x) \| p(z^x) \right] + D_{KL} \left[ q(z^y | y) \| p(z^y) \right] \right] && (20) \\
& \quad - \mathbb{E}_{p_D(x, y)} \left[ D_{KL} \left[ q(z^s | x, y) \| r^y(z^s | y) \right] + D_{KL} \left[ q(z^s | x, y) \| r^x(z^s | x) \right] \right] \\
& \quad + H(X) + H(Y).
\end{aligned}
$$

### A.3.2 Merging ELBO and information preference

Note that Eq. (19) and Eq. (20) coexist in ELBO as reconstruction and KL regularization terms (on exclusive representation) respectively. Thus, the final objective is as follows:

$$
\max_{p,q} \mathbb{E}_{q(z^x, z^s, z^y, x, y)} \left[ \log \frac{p(x, y, z^x, z^s, z^y)}{q(z^x, z^s, z^y | x, y)} \right] + \lambda \left( 2 \cdot I(X; Y; Z^S) - I(Z^X; Z^S) - I(Z^Y; Z^S) \right)
$$

$$
\geq \max_{p,q,r} (1 + \lambda) \cdot \mathbb{E}_{p_D(x,y)} \left[ \mathbb{E}_{q(z^x|x)q(z^s|x,y)} [\log p(x|z^x, z^s)] + \mathbb{E}_{q(z^y|y)q(z^s|x,y)} [\log p(y|z^y, z^s)] \right]
$$

$$
- (1 + \lambda) \cdot \mathbb{E}_{p_D(x,y)} \left[ D_{KL} \left[ q(z^x|x) \| p(z^x) \right] + D_{KL} \left[ q(z^y|y) | p(z^y) \right] \right]
$$

$$
- \mathbb{E}_{p_D(x,y)} \left[ D_{KL} \left[ q(z^s|x,y) \| p(z^s) \right] \right]
$$

$$
- \lambda \cdot \mathbb{E}_{p_D(x,y)} \left[ D_{KL} \left[ q(z^s|x,y) \| r^y(z^s|y) \right] + D_{KL} \left[ q(z^s|x,y) \| r^x(z^s|x) \right] \right]
$$

$$
= \max_{p,q,r} (1 + \lambda) \cdot \mathbb{E}_{p_D(x,y)} \left[ ELBO(p, q) \right]
$$

$$
+ \lambda \cdot \mathbb{E}_{p_D(x,y)} \left[ D_{KL} \left[ q(z^s|x,y) \| p(z^s) \right] \right]
$$

$$
- \lambda \cdot \mathbb{E}_{p_D(x,y)} \left[ D_{KL} \left[ q(z^s|x,y) \| r^y(z^s|y) \right] + D_{KL} \left[ q(z^s|x,y) \| r^x(z^s|x) \right] \right].
$$

## B  Additional quantitative results

### B.1  Sample quality evaluation

| Translation | pix2pix [23] | CdDN [12] | IIAE |
|---|---|---|---|
| $X \rightarrow Y$ | $0.24987 \pm 0.00780$ | $0.23517 \pm 0.00799$ | $\mathbf{0.21478} \pm 0.00844$ |
| $Y \rightarrow X$ | $0.21524 \pm 0.00704$ | $0.19295 \pm 0.00687$ | $\mathbf{0.15277} \pm 0.00774$ |

Table 4: Sample quality evaluation on the Cars (bimodal) dataset.

We report quantitative evaluation on the quality of samples in table 4. We followed the exact experimental setting for the Cars dataset as in [12], except we use freshly generated training data (the data from [12] was unavailable) and the updated version of the evaluation metric LPIPS. Thus, the numbers here do not exactly match those in [12]. The results show that the sample quality of IIAE clearly exceeds the quality of GAN-based methods.

### B.2  Additional notes on table 2

Regarding the numbers from the Facades dataset in table 2, they are different from [12] since we used test set rather than the validation set (stated in the footnote). The table 5 shows the result on the validation set, which matches the numbers in [12].

| Facades(Val) | BicycleGAN [47] | CdDN [12] | IIAE |
|---|---|---|---|
| $F \rightarrow L$ (%) | - | 95.0 (1.0) | $\mathbf{100.0}$ (1.0) |
| $L \rightarrow F$ (%) | 45.0 | 97.0 (1.0) | $\mathbf{100.0}$ (0.0) |

Table 5: Shared (exclusive) representation based retrieval on validation set in the Facades [41] dataset.

### B.3  Ablation study

| Metric | II | II-MI | ELBO+$\lambda$II | ELBO+$\lambda$(II-MI) |
|---|---|---|---|---|
| mAP | 0.517 | 0.534 | 0.516 | $\mathbf{0.573}$ |
| P@100 | 0.605 | 0.616 | 0.595 | $\mathbf{0.659}$ |

Table 6: Ablation study on ZS-SBIR.

We evaluated the effect of terms in the IIAE objective using the ZS-SBIR dataset. Table 6 summarizes the result. II represents maximizing only the interaction information among $X$, $Y$, and $Z^S$ (Eq. (21)),

whose lower bound is as follows:

$$2 \cdot I(X;Y;Z^S) = I(X;Z^S) + I(Y;Z^S) - I(X;Z^S|Y) - I(Y;Z^S|X) \tag{21}$$

$$\geq \mathbb{E}_{p_D(x,y)} \big[ \, \mathbb{E}_{q(z^s|x,y)} [\log p(x|z^s)] + \mathbb{E}_{q(z^s|x,y)} [\log p(y|z^s)] \, \big]$$
$$- \mathbb{E}_{p_D(x,y)} \big[ \, D_{KL} [q(z^s|x,y)\|r^y(z^s|y)] + D_{KL} [q(z^s|x,y)\|r^x(z^s|x)] \, \big]$$
$$+ H(X) + H(Y). \tag{22}$$

II-MI is the joint information preference of maximizing the interaction information and minimizing mutual information between shared and domain-specific representations simultaneously (Eq. (23)), whose lower bound is Eq. (25).

$$2 \cdot I(X;Y;Z^S) - I(Z^X;Z^S) - I(Z^Y;Z^S) \tag{23}$$
$$= I(X;Z^X,Z^S) + I(Y;Z^Y,Z^S) - I(X;Z^X) - I(Y;Z^Y) - I(X;Z^S|Y) - I(Y;Z^S|X) \tag{24}$$

$$\geq \mathbb{E}_{p_D(x,y)} \big[ \, \mathbb{E}_{q(z^s|x,y)q(z^x|x)} [\log p(x|z^x,z^s)] + \mathbb{E}_{q(z^s|x,y)q(z^y|y)} [\log p(y|z^y,z^s)] \, \big]$$
$$- \mathbb{E}_{p_D(x,y)} \big[ \, D_{KL} [q(z^x|x)\|p(z^x)] + D_{KL} [q(z^y|y)\|p(z^y)] \, \big]$$
$$- \mathbb{E}_{p_D(x,y)} \big[ \, D_{KL} [q(z^s|x,y)\|r^y(z^s|y)] + D_{KL} [q(z^s|x,y)\|r^x(z^s|x)] \, \big]$$
$$+ H(X) + H(Y). \tag{25}$$

Last two columns in table 6 represent taking weighted sum with the ELBO, treating $\lambda = 2$ as the hyperparameter. The final column is the objective of IIAE.

The first two columns imply that augmenting the minimization of the mutual information to the maximization of the interaction information is beneficial. This is because the optimization of Eq. (21) gives $Z^S$ an implicit trade-off between capturing domain-specific information to maximize the first and second terms and emptying domain-specific information to minimize the third and fourth terms in Eq. (21). Thus, encoding the domain-specific information in addition to the shared information can be one of optimal solutions for $Z^S$. On the other hand, optimizing Eq. (23) (or Eq. (24)) eliminates the trade-off since the first and second terms in Eq. (24) allow $Z^S$ to share with $Z^X$ and $Z^Y$ the burden of being informative to $X$ and $Y$. Consequently, the optimal solution of Eq. (23) is that $Z^S$ encodes only the information shared across $X$ and $Y$ while $Z^X$ and $Z^Y$ encode only the domain-specific information.

Finally, the last two columns show that the joint information preference is better suited to ELBO than maximization of the interaction information only and gains further performance improvement.

## C   Visualization

### C.1   Additional samples of cross-domain image translation

#### C.1.1   MNIST-CDCB [12]

We present additional samples of image translation with IIAE in table 7. Furthermore, we generate visual analogies using IIAE which are presented in table 8. For each row, we show the queries (the first and fourth columns), references (the second and fifth columns), and the synthesized images (the third and sixth columns). Queries are sources of shared representation, which is digit identity, whereas references are sources of exclusive representations, which are color variations. Tables 7 and 8 shows that IIAE extracts and preserves both of domain specific and shared representations properly.

| $X \rightarrow Y$ | | | | | $Y \rightarrow X$ | | | | |
|---|---|---|---|---|---|---|---|---|---|
| **Input** | **Outputs w/ different** $z^y$ | | | | **Input** | **Outputs w/ different** $z^x$ | | | |
| x | $z_1^y, z_2^y, z_3^y \sim p(z^y)$ | | | $\mu^y$ | y | $z_1^x, z_2^x, z_3^x \sim p(z^x)$ | | | $\mu^x$ |

Table 7: Additional cross-domain translation results in MNIST-CDCB [12] by IIAE. In MNIST-CDCB, domain-specific factors of variation are color variation in background($X$) and in foreground($Y$) while the common factor is the digit ID.

| X | | | Y | | |
|---|---|---|---|---|---|
| **query** | **reference** | **Output** | **query** | **reference** | **Output** |

Table 8: Visual analogies generated by IIAE, synthesizing shared representation from the query and exclusive representations from the reference in each domain.

### C.1.2 Cars [36]

In this section, we compare IIAE with CdDN [12] with Cars dataset. We present additional samples of image translation with IIAE in table 9 and samples from CdDN in table 10 with the same input images. Furthermore, we generate visual analogies using IIAE which are presented in table 11. To achieve the result of table 10 without pretrained model not available, we trained CdDN for Cars dataset (the version with 23 different views) using the code and following the hyperparameter settings released by [12]. Tables 9 and 10 shows that IIAE achieves not only better sample quality but also better disentanglement. In each row of table 10, The content of the given car exposed dependency on the exclusive representation; The details of car such as shape or color varies depending on if exclusive representation is sampled from its prior distribution (the second, third, fourth and seventh, eighth, ninth columns) or extracted from ground-truth pair (the fifth and tenth columns). In table 11, we show the query (the first column), 2 references with different orientation (the sencond and fourth columns), and two synthesized images (the third and fifth columns). Queries are sources of shared representation, which is car identity, whereas references are sources of exclusive representations, which are variations in orientation. We present only the analogy of $Y$ domain, since factors of variation only exists in $Y$.

| $X \rightarrow Y$ | | | | $Y \rightarrow X$ | | | |
|---|---|---|---|---|---|---|---|
| **Input** | **Outputs w/ different** $z^y$ | | | **Input** | **Outputs w/ different** $z^x$ | | |
| x | $z_1^y, z_2^y, z_3^y \sim p(z^y)$ | | $\mu^y$ | y | $z_1^x, z_2^x, z_3^x \sim p(z^x)$ | | $\mu^x$ |

Table 9: Additional cross-domain translation results in Cars [36] generated by IIAE. In Cars, domain-specific factors only exists in $Y$, views in 23 different degrees, while the shared factor is the identity of car.

| $X \rightarrow Y$ | | | | $Y \rightarrow X$ | | | |
|---|---|---|---|---|---|---|---|
| **Input** | **Outputs w/ different** $z^y$ | | | **Input** | **Outputs w/ different** $z^x$ | | |
| x | $z_1^y, z_2^y, z_3^y \sim p(z^y)$ | | $\mu^y$ | y | $z_1^x, z_2^x, z_3^x \sim p(z^x)$ | | $\mu^x$ |

Table 10: Cross-domain translation results in Cars [36] generated by CdDN [12].

|  | | $Y$ | | |
| **query** | **reference1** | **Output1** | **reference2** | **Output2** |
|  |  |  |  |  |
|  |  |  |  |  |
|  |  |  |  |  |
|  |  |  |  |  |
|  |  |  |  |  |
|  |  |  |  |  |
|  |  |  |  |  |

Table 11: Visual analogies generated by IIAE, synthesizing shared representation from the query and exclusive representations from the reference in $Y$ domain.

## C.2 Cross-domain retrieval

In this section, we visualize the top-3 retrieved images of the cross-domain retrieval task in Facades [41] and Maps [17] datasets. For each query image, we classify the result as a success only when the ground truth pair of the query is retrieved as the closest one (top-1), failure otherwise. Although IIAE performs close to perfect in this task, there exist a few of failure cases which we present here as well.

### C.2.1 Maps [17]

Figure 5: Successful examples of cross-domain retrieval (Top-3) in Maps using IIAE.

Figure 6: Unsuccessful examples of cross-domain retrieval (Top-3) in Maps using IIAE.

## C.2.2 Facades [41]

Query(F)  GT(L)          F → L            Query(L)  GT(F)          L → F

Figure 7: Successful examples of cross-domain retrieval (Top-3) in Facades using IIAE.

Query(F)  GT(L)          F → L            Query(L)  GT(F)          L → F

Figure 8: All unsuccessful cases of cross-domain retrieval (Top-3) in Facades using IIAE.

## C.3 ZS-SBIR

Figure 9: Top-10 ZS-SBIR samples from IIAE on the Sketchy Extended dataset. Sketches in the leftmost columns are queries and rest are retrieved candidates (Top-1 to 10 from the left to right). Green checkmark indicates correct retrieval, whereas red crossmark indicates wrong retrieval.

# D  Implementation details

Here we describe the network architectures of our implementation. For any dataset, every convolutional layer or fully connected layer in encoders is followed by batch normalization (BN) and LeakyReLU with slope 0.2, except the last layers of distribution encoders $q(z^x|x)$, $q(z^y|y)$, $r(z^s|x)$, $r(z^s|y)$, and $q(z^s|x,y)$. The output of those last layers are means and log variances. Note that feature extractors (FE) of $r(z^s|x)$ and $r(z^s|y)$ are shared with $q(z^s|x,y)$. Our implementation is publicly available.[2]

## D.1  Network Architecture for MNIST-CDCB [12], Cars [36], Maps [17], and Facades[41]

| Encoder | $q(z^x|x)$ or $q(z^y|y)$ | FE |
|---|---|---|
| Input | 256 x 256 x 3 image | 256 x 256 x 3 image |
| Layer1 | 4x4 Conv. w/ stride 2 and 32 filters | 4x4 Conv. w/ stride 2 and 32 filters |
| Layer2 | 4x4 Conv. w/ stride 2 and 64 filters | 4x4 Conv. w/ stride 2 and 64 filters |
| Layer3 | 4x4 Conv. w/ stride 2 and 128 filters | 4x4 Conv. w/ stride 2 and 128 filters |
| Layer4 | 4x4 Conv. w/ stride 2 and 256 filters | 4x4 Conv. w/ stride 2 and 256 filters |
| Layer5 | FC. 16 | - |

| Encoder | $r(z^s|x)$ or $r(z^s|y)$ | $q(z^s|x,y)$ |
|---|---|---|
| Input | FE(x) or FE(y) | [FE(x) ; FE(y)] |
| Layer1 | 4x4 Conv. w/ 256 filters | 4x4 Conv. w/ 256 filters |
| Layer2 | FC. 256 | FC. 256 |
| Layer3 | FC. 256 | FC. 256 |

| Decoder | $p(x|z^x,z^s)$ or $p(y|z^y,z^s)$ |
|---|---|
| Input | $[z^x;z^s]$ or $[z^y;z^s]$ |
| Layer1 | FC. 262,144, BN, Dropout(0.5), ReLU |
| Layer2 | 4x4 Deconv. w/ stride 1/2 and 512 filters, BN, Dropout(0.5), ReLU |
| Layer3 | 4x4 Deconv. w/ stride 1/2 and 256 filters, BN, Dropout(0.5), ReLU |
| Layer4 | 4x4 Deconv. w/ stride 1/2 and 128 filters, BN, ReLU |
| Layer5 | 4x4 Deconv. w/ stride 1/2 and 64 filters, BN, ReLU |
| Layer6 | 4x4 Deconv. w/ stride 1/2 and 3 filters and Tanh activation |

Note that the last two fully connected layers in shared representation encoders ($r(z^s|x)$, $r(z^s|y)$, and $q(z^s|x,y)$) and the first fully connected layer in decoders are only applied to Cars [36] dataset.

## D.2  Network architecture for Sketchy Extended [29, 37] (ZS-SBIR)

| Encoder | $q(z^x|x)$ or $q(z^y|y)$ | FE |
|---|---|---|
| Input | 512 image feature | 512 image feature |
| Layer1 | FC. 512 | FC. 512 |
| Layer2 | FC. 256 | - |
| Layer5 | FC. 128 | - |

| Encoder | $r(z^s|x)$ or $r(z^s|y)$ | $q(z^s|x,y)$ | Decoder | $p(x|z^x,z^s)$ or $p(y|z^y,z^s)$ |
|---|---|---|---|---|
| Input | FE(x) or FE(y) | [FE(x) ; FE(y)] | Input | $[z^x;z^s]$ or $[z^y;z^s]$ |
| Layer1 | FC. 256 | FC. 512 | Layer1 | FC. 128 |
| Layer2 | FC. 128 | FC. 128 | Layer2 | FC. 512 |

## D.3 Hyperparameters

| Hyper-parameters | Datasets | | | | |
|---|---|---|---|---|---|
| | MNIST-CDCB | Cars | Facades | Maps | Sketchy Extended |
| Learning rate | 0.0002 | | | | |
| Lambda | 5 | 50 | 1,000 | 50 | 2 |
| Reconstruction weight | 1,000 | | | 20,000 | 10 |

## Footnotes

[2] https://github.com/gr8joo/IIAE