[Reviews · NeurIPS 2020]

Review 1

Summary and Contributions: This paper presents a generative model called Interaction Information Auto-Encoder (IIAE) for solving the cross-domain disentanglement problem. Based on information theory, the authors derive an objective function that simultaneously learns domain-invariant and domain-specific representations. Experiments show that IIAE can be used to solve the tasks of cross-domain image translation and image retrieval.

Strengths: The paper focuses on solving an interesting problem of disentangling domain-invariant and domain-specific representations. The objective function is derived based on information theory. The experimental results on Zero-shot sketch based image retrieval (ZS-SBIR) look promising.

Weaknesses: The motivation of this paper does not look very strong. The cross-domain image translation problem can be solved with the existing model CdDN. For the image retrieval task, only the representation $z^s$ is used; but $z^x$ and $z^y$ are not. Therefore, for this task, we might only need a model that learns $z^s$ properly. It is not clear how simultaneously learning $z^x$ and $z^y$ is useful in solving a real-world problem. The paper tries to solve a disentangling problem. But there is no quantitative evidence showing that the learned representations are disentangled. There is also no evidence showing that a disentangled representation helps the downstream tasks. Note that the model might still perform well on the image retrieval tasks, even if $z^s$ is highly dependent on $z^x$ and $z^y$.

Correctness: The derivation for the objective function looks correct. The setups for the experiments look reasonable.

Clarity: In general, this paper is well written. The assumptions about the data are not clearly mentioned. It looks like the paper assumes that for each $x \in X$ there are multiple $y \in X$ that can be paired with x, and vice versa. If the pairing is unique, it seemsthat we only need $Z^S$ to explain both $x$ and $y$ and we do not need $Z^X$ and $Z^Y$ as latent variables. .

Relation to Prior Work: The authors do not mension how the proposed method differs from CdDN [12] and its follow-ups [26, 28, 35, 44]. It looks like CdDN can also generate results as shown in Table 1. It would be best if the authors can provide quantitative comparisons with cdDN, pix2pix and BicycleGAN, as shown in [12].

Reproducibility: Yes

Additional Feedback: Thanks for the clarification and additional evaluations provided in the author feedback.


Review 2

Summary and Contributions: The paper proposes an information-theoretic regularization for cross-domain VAE to learn decomposed and disentangled latent representations. The proposed regularization simultaneously (a) minimizes the mutual information between domain-specific and shared representations to enforce disentanglement, and (b) maximizes the interaction information (generalized mutual information) among the domain data spaces and their shared latent representation. The intractability of individual terms is addressed by deriving tractable bounds that can be optimized, which entails inference domain-specific models that estimate domain-specific and shared posteriors. The effectiveness of the proposed model is showcased on image-to-image translation, image retrieval, and zero-shot retrieval.

Strengths: The proposed regularization framework has principled grounds in information theory, which is novel in the context of cross-domain VAEs. Results demonstrate state-of-the-art performance in image-to-image translation, image retrieval, and zero-shot retrieval. The work addresses cross-domain learning representations that are disentangled into domain-invariant and domain-specific representations, a relevant area to the NeurIPS community with several applications that are also relevant, e.g., domain transfer and content-based retrieval.

Weaknesses: The formulation relies on paired data scenarios with no argument/discussion given for semi-supervised learning or unpaired scenarios. It is not clear how the proposed regularization framework scales to problems that entail more than two domains. The tightness of the proposed bound is not analyzed. The use of Gaussian prior can lead to posterior collapse, affecting the accuracy and diversity of cross-domain models. Results do not assess sample generation quality for the proposed model compared with SOTA multi-view/cross-domain learning generative models.

Correctness: As the proposed method involves augmenting the ELBO with regularizers, there is no theoretical or empirical arguments/analyses for the tightness of the proposed bounds. Furthermore, it is known that regularizing ELBOs could induce latent pockets (submanifolds in the latent space), which impacts sample generation. Domains could follow different generative processes, how would the proposed model support/allow many-to-many mappings? Latent Gaussian priors could lead to strong regularization or posterior collapse, which could limit the diversity of the cross-domain generative models. How robust is the proposed regularizer to the choice of the prior?

Clarity: The paper is well written and organized with proofs and visual results given in the supplemental material. In line 122, should the q(x|z^x,z^s) be the decoder? Are the feature extractor networks shared across domains?

Relation to Prior Work: The paper include enough coverage of the related work for disentangled and invariant representation, but it lacks related works that focus on sample generation across domains.

Reproducibility: Yes

Additional Feedback:


Review 3

Summary and Contributions: This paper suggests a cross-domain disentanglement model based on variational Auto-Encoder. They assume two domains exhibit domain-specific factors of variations while sharing some common factors of variations, which is achieved based on information theory. From the shared information separated with the exclusive features, they could obtain image translation between two different domains and image retrieval from another domain.

Strengths: To control the mutual information aforementioned tractably, they derive proper lower or upper bound, which is fully explained in their supplement as well as the main paper. By combining it with ELBO loss, they provide a new joint objective which makes their model not require adversarial training or gradient reversal layers to train. It is reasonable approach to have disentangled domain-specific and domain-invariant latent space.

Weaknesses: Having separated network for the shared and exclusive features is widely used in cross-domain problem. For the disentanglement, there have been many research that apply the concept of Total Correlation such as FactorVAE or beta-TCVAE. Thus, the novelty is not considerable. In order to argue the effectiveness of the application of mutual information into disentanglement model, the ablation study is necessary. Though they provide results about how the domain-specific representation makes poor cross-domain retrieval in Table 2, it is insufficient for the ablation study. They need to figure out how each regularizer has an impact on the performance by removing it one-by-one. In other words, is it critical not to minimize I(Z^x; Z^s) or not to maximize I(X;Y;Z^s)? But it seems it is not simple and explicit to conduct this experiment in the current objective function which is combined with ELBO and summarized all together with those two regularizers.

Correctness: Based on the information theory, their argument of controlling mutual information is well proved. The derived equations are well-combined for the joint objective with ELBO. Their experimental datasets and setup are commonly used for cross-domain problems, which is enough to ground their claims.

Clarity: Overall, the writing is good to understand. Their main idea and contribution are clearly clarified. The description about the experimental setup is well-defined. However, it would be better to deal with the variational distribution r^x or r^y more carefully. In the line 133 of the main paper, it is written "fits r^y(z^s|y) to q(z^s|y)". But in the line 459 of the supplement, "fits r_s(z^s|y) to q(z^s|y)", which makes confusion. Also, in the line 205 of the main paper, it seems "mu ~ r(*)" should have been "mu^y ~ q(zYy|y)"" seeing line 197. r(*) does not make sense since it samples shared factors.

Relation to Prior Work: They introduced related works in two streams: In variant representation and Disentangled representation. Though it is adequate direction, the details about related works are deficient. Especially, since CdDN is used as their baseline, they need to explain how CdDN designed their model in detail and how the author's work is different with it or improved from it. Also, it would be better to elaborate a bit more some of the cross-domain disentanglement works with unpaired dataset.

Reproducibility: Yes

Additional Feedback: More valid baseline results in 4.2.1 Cross-domain retrieval are required. In Table 9 in the Supplementary, DRIT cannot be a fair baseline since it is trained with "unpaired" dataset in oppose to other methods which use paired data. It is required to make sure of training DRIT with paired dataset or have another baseline such as BicycleGAN which is based on paired dataset. Also, CdDN reported different results for Facade in its paper. F->L : 95 becomes 84.9 and L->F: 97 becomes 89.6 in this paper. If the results of this paper are compared with the originally reported figures in CdDN, it is hard to say there is a significant outperformance in cross-domain retrieval.

[Author Response · NeurIPS 2020]

**R1: Motivation**  Learning disentangled representations in cross-domains is useful for real-world problems such as image translation (demonstrated in the paper) and language translation. CdDN is one of the recent promising work on the cross-domain disentanglement task. However, it is a GAN-based architecture with the gradient reversal layer, which is not ideal for training in our opinion. Our work, IIAE, has a much simpler architecture with a more direct training scheme. The advantage of IIAE can be appreciated by the quality of results, compared to CdDN in Tables 6 and 7.

**R1: Learned representations for image retrieval**  We think we can still check whether the learned representations are disentangled in the image retrieval task. In Table 2, we also report the retrieval accuracy using the exclusive representation (numbers in the parenthesis), which is closed to a random guess (100/N%) showing the successful disentanglement. In contrast, the results from CdDN are noticeably high or low, suggesting that it was relatively unsuccessful in disentangling the representations. Please see below for additional experiments.

**R1: Clarification on the data**  We assume that the pairing is not unique, as in the CdDN paper.

**R1:  Quantitative evaluation**  We report quantitative evalua-tion on the quality of samples, as request by R1. We followed the exact experimental setting

| Translation | pix2pix [23] | CdDN [12] | IIAE |
|---|---|---|---|
| $X \to Y$ | $0.24987 \pm 0.00780$ | $0.23517 \pm 0.00799$ | $\mathbf{0.21478} \pm 0.00844$ |
| $Y \to X$ | $0.21524 \pm 0.00704$ | $0.19295 \pm 0.00687$ | $\mathbf{0.15277} \pm 0.00774$ |

for the Cars dataset as in [12], except we use freshly generated training data (the data from [12] was unavailable) and the updated version of the evaluation metric LPIPS. Thus, please understand that the numbers here do not exactly match those in [12]. The results show that the sample quality of IIAE clearly exceeds the quality of GAN-based methods.

**R2: Limitations**  For Cars and Sketchy datasets, we randomly paired samples within categories, which is a straightfor-ward way to use our method for unpaired samples. Extending to semi-supervised learning tasks and scaling to multiple domains remain as future work. As for the quantitative comparison with SOTA, please see our response to R1 above.

**R2: Correctness of the lower bound optimization**  Our training objective is II minus MI, whose lower bound (ELBO with regularization) is derived taking the standard steps for obtaining variational lower bounds. Thus, this lower bound has the same tightness property as ELBO and VIB.

Please note that our approach is independent of the choice of the prior, although all the experiments used the Gaussian prior for the simplicity in the implementation. Yet, in all of our experiments, we were not able to observe any of the sample diversity issues even under the Gaussian prior. Please refer to Table 1, 4, and 6 demonstrating that our method generates diverse samples depending on $z^x$ and $z^y$.

**R3: Comparison to TC regularization**  FactorVAE and $\beta$-TCVAE are for the *single-domain* disentanglement task, which minimize the total correlation (TC) among all dimensions of the latent variable to make them independent. The cross-domain disentanglement aims to decompose domain-specific and domain-invariant factors of variation into three latent variables (one shared and two exclusives for two domains). Minimizing TC is not directly applicable to cross-domain disentanglement. To the best of our knowledge, our work is the first to introduce the notion of interaction information for the cross-domain disentanglement task.

**R3:  Ablation Study**  We conducted ablation study on the effect of terms in the IIAE objective, using the ZS-SBIR dataset. II represents optimiz-ing interaction information only, and II-MI is the objective in (11). Last two columns represent taking

| Metric | II | II-MI | ELBO+$\lambda$II | ELBO+$\lambda$(II-MI) |
|---|---|---|---|---|
| mAP | 0.517 | 0.534 | 0.516 | **0.573** |
| P@100 | 0.605 | 0.616 | 0.595 | **0.659** |

weighted sum with the ELBO, treating $\lambda = 2$ as the hyperparameter. The final column is the objective of IIAE. Comparing to Table 3, all settings significantly outperform SOTA, and shows that subtracting MI from II always help.

**R3: Additional comments on Table 9**  Please note that we re-trained DRIT using the *paired* data in order to make a fair comparison (stated in the text), via minor modification to the author's code to take advantage of the paired data. Regarding the numbers from the

| Facades(Val) | BicycleGAN | CdDN | IIAE |
|---|---|---|---|
| $F \to L$ (%) | - | 95.0 (1.0) | **100.0** (1.0) |
| $L \to F$ (%) | 45.0 | 97.0 (1.0) | **100.0** (0.0) |

Facades dataset, they are different from the original paper since we used test set rather than the validation set (stated in the footnote). The table on the right shows the result on the validation set, which matches the numbers in the original paper. Finally, thank you very much for catching typos, which will be fixed in the final version of the paper.

[Meta-Review · NeurIPS 2020]

The paper introduces a model for disentanglement of domain-specific and domain-invariant representation. Despite its limitation (e.g., only works with paired data), the strength of the paper is in its principled information-theoretic treatment of the problem. The author's response does help clarify some of the concerns in the original reviews. My recommendation is to accept, however, I would strongly recommend the authors to revise the papers to include the materials provided in the rebuttal.